# Elucidating the Neurobiologic Etiology of Comorbid PTSD and Substance Use Disorders

**DOI:** 10.3390/brainsci12091166

**Published:** 2022-08-31

**Authors:** Jesse D. Hinckley, Carla Kmett Danielson

**Affiliations:** 1Division of Addiction Science, Treatment & Prevention, Department of Psychiatry, University of Colorado School of Medicine, 1890 N Revere Court, MS-F570, Aurora, CO 80045, USA; 2National Crime Victims Research & Treatment Center, Department of Psychiatry & Behavioral Sciences, Medical University of South Carolina, 67 President Street, MSC 861, Charleston, SC 29425, USA

**Keywords:** PTSD, substance use disorders, adolescence, amygdala, prefrontal cortex, mesolimbic dopamine system, noradrenergic system, hypothalamic-pituitary-adrenal axis

## Abstract

Early childhood maltreatment and other traumatic event experiences (“trauma”) are common among youth, including those with substance use problems including substance use disorders (SUD). Particularly, interpersonal violence is associated with high rates of comorbidity between posttraumatic stress disorder (PTSD) and SUD, and these comorbid disorders exhibit high levels of overlapping symptomatology. Theoretical models proposed to explain the bidirectional relationship between PTSD and SUD include the self-medication hypothesis and susceptibility hypothesis. In this article, we explore neurobiologic changes associated with trauma, PTSD, and SUD that underly dysregulated stress response. Examining lessons learned from recent translational and clinical research, we propose that further elucidating the neurobiologic etiology of comorbid PTSD and SUD will require a collaborative, interdisciplinary approach, including the integration of preclinical and clinical studies, exploration of biologic markers in clinical studies, and accumulation of larger studies and longitudinal studies with the power to study PTSD and SUD. Such research can transform the field and ultimately reduce high rates and costly impairment of co-occurring PTSD and SUD across the lifespan.

## 1. Introduction

Post-traumatic stress disorder (PTSD) results from exposure to a potentially traumatic event, either by directly experiencing or witnessing the event or by learning about an event that occurred to a close family member or friend [1]. Approximately two-thirds of children experience at least one significant traumatic event by age 16 years-old [2,3]. Early childhood maltreatment, including interpersonal trauma, is one of the most common causes of PTSD in youth [4]. In 2020, there were 3.9 million reports of childhood maltreatment to Child Protective Services in the U.S. involving 7.1 million youth (28.9 reports per 1000 children) [5]. Further, data from the National Child Abuse and Neglect Data System Child File show that 1 in 8 children will have a confirmed report of child maltreatment [6]. However, due to underreporting and difficulty following up, most studies and reports grossly underestimate the prevalence of childhood maltreatment and PTSD that extends from such trauma exposure.

The diagnosis of PTSD requires a constellation of symptoms and behaviors that result in clinically significant distress or impairment and includes: (1) recurrent intrusive symptoms; (2) persistent avoidance of stimuli associated with the traumatic event; (3) negative alterations in mood and cognition; and (4) alterations in arousal and reactivity associated with the traumatic event [1]. Notably, many youths do not have insight into avoidant behaviors, cognitive or affective changes, and hyperarousal or other reactivity changes that result from the trauma. As a result, while children are more likely to be diagnosed with PTSD after a traumatic experience than adults, many traumatized children may not be diagnosed with PTSD and may manifest other mental health disorders [4]. This diagnostic challenge further complicates characterizing the mental health impact of trauma in youth and investigating the mechanistic etiology of trauma-related psychopathology.

Specific to the adolescents, traumatic experiences are common among young people with substance use disorders (SUD) [7]. Further, youth with PTSD are up to 14 times more likely to have a comorbid SUD than youth without PTSD. In particular, youth exposed to interpersonal violence, such as child sexual abuse, physical abuse, or witnessed violence, are more likely to use substances including cannabis, cocaine, opioids, and other drugs [8]. In fact, interpersonal violence exposure is a risk factor for substance use problems in adolescence and adulthood and the development of SUD [4,9], even after controlling for family background and parental psychopathology [10]. Interpersonal violence exposure also serves as a strong risk factor for comorbid diagnoses of PTSD and alcohol use disorder than either diagnosis individually [11]. Thus, traumatic experiences, particularly interpersonal traumas, are important risk factors for the development of comorbid trauma and stress disorders and SUD. The presence of problematic substance use also increases the risk of experiencing potentially traumatic events, reinforcing a cycle of exposure to trauma and continued substance use problems among victimized youth [12].

In adolescents, 24–30% of youth with PTSD have comorbid SUD [7]. Similarly, 13–25% of youth with a primary SUD have a diagnosis of PTSD. The estimates of comorbid PTSD and SUD among adults are even higher, ranging from approximately 30–50% and up to 85% among treatment-seeking individuals [12]. Importantly, it is likely that trauma exposure, PTSD, and SUD are underdiagnosed and untreated in youth. This is a multifactorial problem, with contributors including: (1) absent or inadequate screening for a history of potentially traumatic events and substance use problems in clinical and research settings; (2) a lack of adequate training in the nuances of trauma assessment and how PTSD symptoms present in youth; (3) limited treatment seeking among youth with co-occurring PTSD and SUD and their caregivers; and (4) the broader compartmentalization of the mental health and substance use fields, evidenced in funding, education and training, licensure, etc. Together, these problems contribute to challenges in the study of comorbid PTSD and SUD, including prevalence, etiology, assessment, treatment, and prevention [13].

Herein, we discuss the overlapping clinical symptomatology and common neurobiologic etiology of comorbid PTSD and SUD. We offer insights into the importance of evaluating exposure to both trauma and substance use in research to inform future directions in research, assessment, and treatment in this field and to ultimately reduce high rates and costly impairment of co-occurring PTSD and SUD across the lifespan.

## 2. Overlapping Clinical Symptomatology

Clinical experience and research have highlighted a strong, bidirectional relationship between PTSD and SUD. Both PTSD and SUD for various substances also demonstrate overlapping symptomatology. For example, withdrawal from alcohol is characterized by anxiety, irritability, sleep disturbances, and exaggerated startle response. Cocaine intoxication and withdrawal are characterized by hypervigilance, paranoia, anxiety, sleep disturbances, and mood problems. These signs and symptoms of alcohol withdrawal and cocaine intoxication or withdrawal are also core features of PTSD, particularly alterations in arousal and reactivity: irritable behavior, hypervigilance, exaggerated startle response, problems with concentration, and sleep disturbance [1]. Further, the co-occurrence of these disorders exacerbates symptoms of both PTSD and SUD more so than if either disorder presented without comorbidity [14]. Multiple theoretical models have been proposed to explain the significant PTSD-SUD connection. One example is the susceptibility hypothesis, which postulates that substance use increases the likelihood of trauma exposure (e.g., increased vulnerability to trauma events in situations involving heavy alcohol use), which, in turn, is associated with increased risk of developing PTSD after traumatic experiences [12,15].

Beyond substance use serving as a risk factor for later traumatic event exposure, another example stems from recent epidemiological studies positing that PTSD often develops prior to problematic substance use [16]. Specifically, the “self-medication hypothesis” [17] is a negative reinforcement etiologic model that suggests when youth experience trauma-related distress, the youth is primarily motivated to engage in strategies, such as substance use, to avoid or decrease the distress and negative affective state. Each time the youth uses substances and the distress is temporarily “relieved,” it negatively reinforces the substance use behavior and increases the likelihood that the youth will use substances the next time they face a distressing trauma cue. Further extending from the self-medication hypothesis, problematic substance use may also exacerbate PTSD symptoms of avoidant behaviors (e.g., using marijuana to avoid being fully present in situation that serve as trauma reminders) and negative alterations in cognition and mood associated with traumatic events.

As outlined by Volkow and Koob [18,19], a three-stage cycle of addiction vulnerability may explain how self-medication can adversely exacerbate PTSD symptoms and substance use problems. In the first stage, binge patterns of substance use and intoxication result in activation of neural circuits that mediate incentive salience and reward networks. During withdrawal (second stage), loss of reward and activation of stress systems results in a negative affective state. Many of the symptoms of PTSD and substance withdrawal overlap, which may further exacerbate this negative affective state. Subsequently, in the third stage, individuals develop preoccupation with and anticipation of substance use. Dysregulation of the prefrontal cortex (PFC) contributes to increased impulsivity and craving, ultimately increasing the risk for continued binge use and intoxication. Despite the significant overlapping symptomatology between PTSD and SUD and the theoretical models describing their bidirectional relation, additional research is still needed to fully understand how these disorders co-occur. Research examining shared neurobiological etiology of PTSD and SUD also may provide insight into the PTSD-SUD connection.

## 3. Common Neurobiologic Etiology

In addition to overlapping clinical symptomatology, comorbid PTSD and SUD share common biological etiologic factors, such as shared neural substrates and pathways [20] and an altered stress response system. This research unveils contenders for potential prevention and treatment targets for comorbid PTSD and SUD in adolescents [21]. With regard to common neural substrates and pathways altered by trauma and substance use, much of what is known about the shared neurobiologic etiology of PTSD and SUD has been extrapolated from the study of individuals with PTSD and alcohol use disorder [4,12,14,22]. Functional neuroimaging studies have identified changes associated with both PTSD and alcohol use, including hyperactive amygdala and hypoactive ventromedial PFC [14,23]. The amygdala mediates processes essential to comorbid PTSD and SUD, including hyperarousal and fear conditioning and learning. Alterations in amygdala functioning subsequently result in avoidant and drug-seeking behaviors [24]. Repetitive substance use induces long-lasting associative memories of environmental cues and substance use. These synaptic connections are subsequently reinforced by ongoing substance use. It is hypothesized that these changes in the amygdala underly fear reinstatement and increased risk of relapse to substance use when triggered by a cue or stressor [23].

Neuroimaging studies indeed support the centrality of amygdalar changes in PTSD and SUD. Hyperactivity of the amygdala is one of the most consistent neurobiologic changes observed in PTSD [25]. Lower amygdala volume has been shown to be quantitatively associated with cumulative stress exposure [26], as well as inversely correlated with alcohol craving and drinking [14,27]. Preclinical models also underscore the importance of amygdala function in maintaining substance use behaviors. Recently, Rich et al. demonstrated that activating calcineurin in the amygdala can reduce cue-induced reinstatement and lower the risk of relapse, suggesting these adaptive processes are reversible [28]. Combined, these studies highlight the amygdala as potential target for future PTSD-SUD studies with young people.

Another region of the brain implicated in PTSD and SUD is the ventromedial PFC. Exposure to early life stress is associated with several changes in the PFC [29]. In addition, hypoactivity of PFC is correlated with duration of PTSD symptoms in youth [25]. These changes may underly distress, regressive behavior, anxiety, negative affect, substance use, and PTSD [29]. Multiple studies have also revealed that PFC hypoactivity is associated with executive dysfunction and alcohol craving and predicts binge drinking and higher rates of relapse in response to stress [14,30,31,32]. Additionally, PFC projections to amygdala are thought to mediate fear conditioning (a process impacted by PTSD), drug seeking behavior, and extinction learning [23,33]. These hypoactive PFC-amygdala projections result in hyperactivity of the amygdala and subsequently abnormal fear conditioning and drug-seeking behavior [14].

Most of the aforementioned studies of the amygdala and PFC have been conducted exclusively in PTSD or substance-specific exposures. These studies typically have relatively small sample sizes and use a variety of imaging and analytical methods. As a result, neuroimaging results are heterogenous and may appear inconsistent. However, the aggregation of these findings identifies common regions of interest and proposes a mechanistic link between PTSD and SUD mediated by the amygdala and PFC.

PTSD and SUD are also both characterized by hyperarousal and maladaptive responsivity to stress. Three neurotransmission systems have been identified that commonly underly PTSD and SUD and are fundamental to brain development. These include the mesolimbic dopamine system, noradrenergic system, and hypothalamic-pituitary-adrenal (HPA) axis [16,22]. Review of these signaling systems highlights the importance of arousal and regulatory systems in the co-occurrence of PTSD and SUD [34].

Research has shown that childhood adversity results in dysregulation of the mesolimbic dopamine system [14]. Initially, differences in dopaminergic signaling may increase susceptibility to development of PTSD following a traumatic stress and subsequent risk of developing problematic substance use [35]. Similarly, alcohol drinking is initially motivated by dopaminergic regulation of positive reinforcement, followed by down-regulation of the mesolimbic dopamine system [18]. These changes subsequently result in aberrant learning, reward deficiency, and anhedonia, predisposing the individual to drug craving and higher risk of relapse [36].

As a result of chronic changes in mesolimbic dopaminergic signaling, individuals with chronic substance use problems may transition from using for the positive effects, such as euphoria, to using to prevent experiencing negative effects (e.g., withdrawal, craving). Considering the three-stage cycle of addiction proposed by Volkow and Koob, these neuroadaptations also result in exacerbation of withdrawal and negative affect [18]. Clinically, this manifests as increased feelings of depression, anxiety, and restlessness. Consequently, the youth may develop a compulsive pattern of consumption to escape dysphoria, which is fundamental to the self-medication hypothesis.

Traumatic stress also activates the locus coeruleus (LC), triggering a norepinephrine-mediated stress response, including “fight-or-flight” [4,14,22,37]. Youth with a history of trauma exposure show increased baseline functioning of the noradrenergic system and enhanced sympathetic nervous system tone, which are positively correlated with intrusive thoughts, avoidance, and hyperarousal [4,14]. Further, markers of noradrenergic activity are elevated within individuals with PTSD and alcohol or opioid withdrawal [22], as well as escalation of alcohol use following early life stress [14].

Few studies to date have shed light on the effect of substance use on acute stress response [38]. Possible mechanisms include altered setpoints and adaptations in neural signaling pathways that result in dysfunctional cue reactivity and maintain drug use motivation and risk of relapse. To explore the neurophysiologic etiology of comorbid PTSD and SUD, Le Dorze et al. developed a preclinical rat model exposed to trauma which develops PTSD-like symptoms [39]. When the noradrenergic system is activated by amphetamine administration, rats exhibited increased reactivity of mesocortical dopaminergic neurons and increased stress response. Thus, noradrenergic sensitization due to drug use and subsequent changes in dopamine signaling may provide a common physiologic basis of PTSD and SUD [39].

During initial trauma exposure, elevated corticotropin releasing hormone (CRH) levels result in hypersecretion of cortisol [40]. Elevated CRH levels in the amygdala may also mediate hyperarousal and increase fear-related behavioral responses [22,41]. Over time, persistent down-regulation of CHR receptors in the anterior pituitary results in lower baseline cortisol levels. This negative feedback loop primes the HPA axis to be hypersensitive to stress, resulting in maladaptive stress response [4]. Higher CRH levels in the hypothalamus have also been associated with withdrawal from cocaine and after alcohol administration [12]. Considering these studies, elevated CRH may mediate the effect of stress, including hyperarousal, on increased substance use. As proposed by Koob, elevated CRH levels in the LC increases norepinephrine turnover, including in the amygdala, which subsequently stimulates the release of CRH in a feedforward loop that progressively increases stress response with repeated stressors [42,43]. Multiple types of substances have been shown to activate the HPA axis and catecholaminergic system, including nicotine, cannabis, cocaine, and alcohol [38]. The consequences of opioid use appears more complicated, with human studies showing downregulation of these systems and animal studies showing activation [38].

Finally, research has shown that exposure to early childhood trauma results in dysregulation of cortisol reactivity [44]. Altered cortisol reactivity may enhance the shared vulnerability for the development of substance use problems, PTSD, and other psychiatric disorders [4]. Thus, targeted SUD prevention among those with altered cortisol reactivity may be a future direction to consider among young people who have experienced early life trauma—or are the offspring of mothers with PTSD [21]. In sum, neuroimaging and lab studies suggest that comorbid PTSD and SUD share common biological etiologic factors, which highlight potential prevention and treatment targets for co-morbid PTSD and SUD in adolescents.

## 4. Lessons from Translational and Clinical Research

To date, many studies have investigated SUD or PTSD in isolation, excluding comorbid problems. However, recent studies continue to highlight the importance of evaluating these often-comorbid disorders within the same model or paradigm. A study by Brooks et al. that examined the association of early life adversity with alcohol use disorder (AUD) in adolescents without psychiatric comorbidity illustrates the importance of considering trauma and substance use in the same model [45]. In this study, the authors found that adolescent AUD was associated with reduced bilateral temporal volumes compared to healthy controls [45]. Regression analyses integrating early life adversity found that higher ratings of childhood trauma were associated with volumetric reductions in the right precentral gyrus and bilateral hippocampus, even after controlling for AUD [45]. The authors conclude that some changes observed in previous studies of AUD may reflect the impact of trauma or other confounders, rather than the effect of alcohol use alone [45].

Cannabis use and PTSD have also both been associated with changes in white matter tracts in the cingulum and anterior thalamic radiations [46]. In a recent study, Yeh et al. sought to investigate the impact of comorbid PTSD and cannabis use on white matter tract integrity by categorizing participants into four groups including trauma-exposed individuals with no PTSD or regular cannabis, individuals with PTSD and no regular cannabis use, trauma-exposed individuals who use cannabis and do not have PTSD, individuals with PTSD and regular cannabis use [46]. The authors found that PTSD was associated with increased fractional anisotropy (FA) in the right anterior thalamic radiation, which correlated with PTSD symptom severity [46]. On the other hand, cannabis use was associated with decreased FA in bilateral anterior thalamic radiata [46]. There was no significant interaction between PTSD and cannabis, indicating that in individuals with PTSD, cannabis use does not further alter PTSD-related alterations in this white matter tract [46].

In another study of cannabis use, Domen et al. investigated white matter integrity longitudinally in individuals with a psychotic disorder, siblings without a psychotic disorder, and healthy controls [47]. Diffusion tensor imaging studies were completed twice at an interval of three years. The authors hypothesized that childhood trauma and cannabis use would show reduced white matter FA over time [47]. When evaluating cannabis use or childhood trauma exposure independent of participant group, the authors found no significant difference in FA [47]. However, when considering group, cannabis and childhood trauma exposure were independently associated with decreases in FA in individuals with psychosis compared to non-psychotic siblings [47].

The evaluation of comorbid cocaine use disorder and PTSD has garnered valuable insights into the overlapping symptomatology and shared neurobiologic etiology of these comorbid disorders [48,49]. Gawrysiak et al. conducted BOLD fMRI scans in 34 treatment-seeking men admitted for cocaine use disorder with and without trauma after 7–10 days of supervised sobriety [50]. Individuals with cocaine use disorder and trauma showed greater functional connectivity between the amygdala and limbic-striatal regions, including nodes of mesolimbic motivational circuitry in the caudate nucleus, putamen, pallidum, and insula, compared to individuals without trauma [50]. The authors propose this heightened state of connectivity may predispose individuals with trauma to increased reactivity to drug-related cues [50]. This, in turn, might explain higher rates of relapse in individuals with comorbid cocaine use disorder and trauma.

These studies highlight the heterogeneity of neuroimaging findings. One possible explanation may be that neural pathways and etiologic factors differ by substance used. Other study factors that may contribute to heterogeneity include participant baseline characteristics, as well as neuroimaging modality and scope of the brain region investigated (i.e., agnostic vs. region-specific). Variability may also be introduced in the analysis strategy, including selection of covariates and sociodemographic factors. Yet, these studies provide examples of strategies to investigate comorbid trauma and SUD and may provide valuable mechanistic insights into these comorbid disorders.

As demonstrated by Gawrysiak et al., identifying neurobiologic changes associated with PTSD and substance use problems may identify potential vulnerabilities for relapse and facilitate the development of targeted interventions. This is especially important in the prevention and treatment of comorbid PTSD and SUD, as these comorbid disorders are associated with earlier onset of SUD, more substance-related problems, poorer treatment adherence and prognosis of both disorders, and poorer overall physical and mental health than having either disorder only [12,14].

Consistent with the proposed cyclical pattern of addiction proposed by Volkow and Koob, clinical studies have shown PTSD symptom severity is positively associated with impulsivity and substance use problems in trauma-exposed individuals [51]. Morris et al. demonstrated impulsive traits may subsequently mediate the association between PTSD and substance use. Similarly, substance use severity and frequency are significantly associated with emotion dysregulation, including difficulty controlling impulsive behaviors [52]. Emotional dysregulation may mediate the behavioral pathway between trauma exposure and problematic substance use [53]. Emotional dysregulation motivates maladaptive behaviors, including problematic substance use, that are often goal-directed to avoid negative affective states. Increased emotional dysregulation is also associated with developing alcohol dependence, more severe alcohol cravings, and frequency of cannabis use [52]. Thus, interventions that target emotional dysregulation may improve both PTSD and SUD [54].

To date, studies of SUD have emphasized the role of PTSD as a significant trigger for ongoing problematic substance use. In a large sample of adolescents receiving substance use treatment (*n* = 20,069), Davis et al. found that ongoing PTSD is a key mechanism for predicting return to substance use [55]. Further, the severity of PTSD symptoms was positively associated with the degree of substance use at the end of treatment [55]. Similarly, an evaluation of justice-involved youth found PTSD symptomatology was not only associated with SUD symptoms, but it may also mediate the association between SUD symptoms and externalizing behaviors [56]. Further, decreasing hyperarousal and impulsivity may be vital to reducing problematic substance use [57].

These findings highlight the importance of treating PTSD to effectively reduce problematic substance use. Few studies have been published investigating the treatment of comorbid disorders in adolescents [58,59]. Danielson et al. recently completed a randomized controlled trial of Risk Reduction through Family Therapy, demonstrating the utility of exposure-based therapy to safely treat PTSD youth with substance use problems [13]. Thus, it is feasible, and vital, to co-manage PTSD and SUD in adolescents.

## 5. Summary and Future Directions

Data from well-designed epidemiological studies show that most people have been exposed to one or more potentially traumatic stressors during their lives. Childhood is no exception, and by age 18 years 1 in 2 youth will experience serious interpersonal violence [60,61]. Exposure to traumatic events levies tremendous mental health burden, including PTSD and SUD, among those who experience traumatic stress, as well as on society at large. PTSD and SUD commonly co-occur in adolescents and adults and appear to have a bidirectional relation, where problematic substance use may predispose individuals to potentially traumatic exposures and individuals with PTSD are more likely to engage in problematic substance use. In this paper, we focused on comorbid PTSD and SUD, with annual financial costs to the US ranging from $460–740 billion [62]. The mental health impact of traumatic stressors appears to be exacerbated by the COVID-19 pandemic, during which we have seen significant increases across a range of behavioral health problems (e.g., opioid overdose) being reported [63,64,65]. In fact, a National Emergency in Child Mental Health has been declared as of October 2021 [66].

These high prevalence rates and costs of PTSD and SUD underscore the significance of and urgency for dedicated attention and resources to the study of their co-occurrence in adolescence, with an eye on curtailing these impacts earlier in the lifespan. While advances in the PTSD-SUD fields have been yielded (as reviewed above), our perspective is that vital empirical questions remain unanswered regarding the neural, genetic, psychophysiological, and behavioral mechanisms that underlie the bidirectional pathway between PTSD and SUD in adolescents. As we call for research dedicated to further elucidate the etiology of PTSD and SUD and their co-occurrence, we recognize the complexity in this area, including shared risk factors, overlapping clinical features, and common neurobiologic pathways. Thus, we offer the following recommendations for future directions in this area.

First, it is critical that an integrated approach is taken–where the mental health and substance use fields come together as a unified field—rather than thinking of these disorders as distinct, compartmentalized entities. Historically, the study and treatment of PTSD and SUD have occurred within silos, often even excluding co-occurrence of the other disorder as part of their research design. Relatively few studies have investigated the complex etiology of comorbid PTSD and SUD within the same model. As a result, the majority of our understanding of these often co-occurring disorders is extrapolated from comparisons between studies of PTSD and studies of substance use. Thus, we propose that to better understand the shared etiology and differences of these disorders, it is vital to consider trauma and substance use within the same model. This may require large, longitudinal studies with less restrictive inclusion criteria [47,67,68].

The Adolescent Brain Cognitive Development (ABCD) study presents a unique resource to characterize the longitudinal impact of comorbid trauma and substance use in the developing brain [69]. The largest longitudinal study of neurodevelopment, ABCD characterizes trauma [70] and substance use [71]. While the study is designed to complete baseline assessments prior to the onset of regular substance use [69], one limitation may be the degree to which the occurrence of early childhood trauma prior to enrollment is fully captured. Nevertheless, strategies to identify the unique and combined effects of these comorbid disorders may inform translational studies to further elucidate the neurobiologic underpinnings of comorbid trauma and stress disorders and SUD [47,67,68].

Second, the characterization of unique and common neurobiologic mechanisms will also benefit from the development of interdisciplinary collaborations that combine clinical and translational studies with preclinical animal studies [14]. Recent reviews highlight the value of integrating translational human studies and preclinical animal models to develop novel insights into the mechanisms of PTSD [16,72,73]. Such an approach has also proven valuable for understanding the neurobiology of addiction [18]. To date, few studies have investigated the impact of trauma and substance use within the same preclinical models. However, a seminal review by Chadwick et al. demonstrated the feasibility of utilizing preclinical models to examine the impact of these comorbid risk factors in subsequent development of psychopathology [74]. Additionally, collaborative approaches that investigate both PTSD and SUD may facilitate the identification of biologic markers that will allow more accurate treatment planning and risk stratification, including risk of relapse.

Third, while some degree of heterogeneity is inherent to ensure representation and generalizability to real world populations, experts should identify guidelines for inclusion and exclusion criteria—such that comparisons and compilations of results can more easily occur across studies. Historically, the wide range of inclusion and exclusion criteria have made it challenging to make conclusions. Differences in the study inclusion and exclusion criteria fall across multiple domains including trauma exposure vs. PTSD, acute substance use vs. chronic substance use, and binge or disordered use vs. naïve or non-disordered use, which can great impact results. For example, the severity of trauma exposure will likely differentially alter biologic systems in response to trauma or to substance use and the impact of subsequent substance use on behaviors. It is therefore not surprising that human studies have found inconsistent results.

Fourth, recruitment and retention of diverse populations across all forms of studies targeting comorbid PTSD and SUD studies are vital to ensure our research is representative and generalizes to the very populations at highest risk for trauma exposure. In a paper published in *Neuroimage* in July 2022, Goldfarb and colleagues presented results from a systematic review that found that only 20 out of 536 articles reported the race and ethnicity demographics of their participants [75]. The authors called for the increased diversity and transparency, which we echo here. We also recommend the inclusion of measurement of forms of trauma that particularly impact diverse populations [76].

Finally, the key role of development needs to be taken into strong consideration when approaching future research in this area. That is, while studies have identified neurobiological differences linked with trauma and SUD in adults, much less is known about the impact of comorbid trauma and substance use on the developing brain during adolescence. Studies to date as reviewed in this paper provide a strong foundation for valuable insights into the unique contributions and shared etiology of comorbid PTSD and SUD and target variables as the next steps in this line of research. Our hope is that research extending from these recommendations and beyond offer a promising pathway to move the needle in improving public health—and ultimately forestall the onset of mental illness, including PTSD and SUD, following exposure to traumatic events and adversity.

Finally, significant barriers remain to effectively engage and retain youth with comorbid disorders into treatment [77]. Ultimately, as we gain increased insights into the shared risk factors and common biologic etiology, these findings will be fundamental to developing improved prevention and treatment options for comorbid trauma and stress disorders and SUD. Elucidation of the neurobiologic etiology of these commonly comorbid disorders may identify new pharmacologic targets. Additionally, translational and clinical studies of comorbid trauma and stress disorders and SUD may identify protective factors that can be enhanced, as well as risk factors that can be mitigated, by psychotherapy. Dissemination and implementation research is vital as youth with comorbid trauma and stress disorders and SUD are among the most undertreated patients due to the difficulty of managing these co-occurring disorders. However, as demonstrated by Danielson et al., it is feasible to improve clinical outcomes through integrated treatment. Thus, additional research is needed to ensure youth with comorbid PTSD and SUD have access to evidence based treatment (RRFT) for these problems. Such advances would prove invaluable to the treatment of comorbid PTSD and SUD and will ultimately reduce the public health burden levied by these problems.

## Data Availability

Not applicable.

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
