# Peer review of "Elucidating the Neurobiologic Etiology of Comorbid PTSD and Substance Use Disorders"

_brainsci, 2022, doi:10.3390/brainsci12091166_

Round 1

Reviewer 1 Report

Dear Authors

Your article is very interesting. But pay attention to the following points:

1. format the text according to the journal's requirements

2. describe the PTSD symptoms and under what conditions the diagnosis is made? 

3. You may report a conclusion that will contain a brief summary of all your data.

4. in my opinion, for the compared article format, the references are very few 

5. Overall, the article is more like a review and does not provide the necessary insight of the authors.

Author Response

Your article is very interesting. But pay attention to the following points:

  1. format the text according to the journal's requirements

Response: We believe the revised manuscript complies with the author guidelines. We apologize for not submitting the original manuscript utilizing the template. We have confirmed section headings and paragraph formatting conform to the template. Front matter sections were verified with the author guidelines. The research manuscript sections are excluded as this article is a “perspective.” Back matter sections have been added and verified with the author guidelines. References utilize a standard formatting and are numbered in the order of utilization.

  1. describe the PTSD symptoms and under what conditions the diagnosis is made? 

Response: The following sentences have been added to the introduction:

Post-traumatic stress disorder (PTSD) results from exposure to a potentially traumatic event, either by directly experiencing or witnessing the vent or learning about the event that occurred to a close family member or friend [1].”

“The diagnosis of PTSD requires a a constellation of symptoms and behaviors that result clinically significant distress or impairment and include 1) intrusive symptoms that are often recurrent or distressing, 2) persistent avoidance of stimuli, 3) negative alterations in mood and cognition, and 4) alterations in arousal and reactivity that associated with the traumatic event [1]. Notably, many youth do not have insight into avoidant behaviors, cognitive or affective changes, and hyperarousal or other reactivity changes that result from the trauma.”

  1. You may report a conclusion that will contain a brief summary of all your data.

Response: We have revised the discussion to review key findings and set the stage for our perspective on how to advance research of comorbid PTSD and substance use disorders.

  1. in my opinion, for the compared article format, the references are very few 

Response: We have updated the article with recent references and added 29 references published since 2017. These references expand the inclusion of more recent findings to support proposed hypotheses and connections and enriches the discussion of clinical research.

  1. Overall, the article is more like a review and does not provide the necessary insight of the authors.

Response: We appreciate this critique. We have revised the manuscript to include more commentary and insight throughout and revised the discussion to highlight our perspective of the current state of science and future directions.

Reviewer 2 Report

This research paper explores the etiological overlap of comorbid PTSD and substance abuse. In particular, the authors focus on the shared vulnerability arising from dysregulation of the systems governing the biological stress response.

The authors review the (primarily neuroimaging) evidence that their comorbidity is a result of hyperactive amygdalar activity and concomitant hypoactivation of the ventromedial PFC. While this review is useful, much of it is quite a bit older, with the majority predating 2015. It would be useful to complement this line of reasoning with a few more recent observations that either support or refute it.  

Their discussion of the three neurotransmitter systems underlying the effects – the mesolimbic dopamine, noradrendergic, and HPA systems – is cogently written and well structured. This is a useful simplification which I believe readers will find valuable.

In terms of their review of the clinical research, the evidence is less convincing. As the authors themselves note, much of the research is in fact contradictory, with studies of concomitant substance use and trauma resulting in increased activity, decreased activity, or no change in the structures under study. While the authors note and attempt to address this heterogeneity, to some extent this section is problematic because it almost undermines the case they have been making throughout regarding their shared etiology and common pathophysiology. Presumably, underlying these disparate findings, it is reasonable that there must be some coherent mechanism? It might be helpful to at least address this problem.

Overall, I found this review well organized, concise, and well-written. In particular, I appreciated the breadth of material covered, as well as their systematic approach to exploring the clinical presentation, neurobiology, and imaging evidence. I also valued their conclusion, that an approach integrating both disorders is necessary to improve outcomes and find better prevention strategies for this highly vulnerable group.

I believe this review will serve as a useful source for understanding comorbid PTSD and substance use, particularly as it is generalized across substance (most previous reviews have been limited to specific substances).

Thank you for the opportunity to review.

Author Response

This research paper explores the etiological overlap of comorbid PTSD and substance abuse. In particular, the authors focus on the shared vulnerability arising from dysregulation of the systems governing the biological stress response.

The authors review the (primarily neuroimaging) evidence that their comorbidity is a result of hyperactive amygdalar activity and concomitant hypoactivation of the ventromedial PFC. While this review is useful, much of it is quite a bit older, with the majority predating 2015. It would be useful to complement this line of reasoning with a few more recent observations that either support or refute it.  

Response: We have updated the article with recent references and added 29 references published since 2017. These references expand the inclusion of more recent findings to support proposed hypotheses and connections and enriches the discussion of clinical research.

Their discussion of the three neurotransmitter systems underlying the effects – the mesolimbic dopamine, noradrendergic, and HPA systems – is cogently written and well structured. This is a useful simplification which I believe readers will find valuable.

In terms of their review of the clinical research, the evidence is less convincing. As the authors themselves note, much of the research is in fact contradictory, with studies of concomitant substance use and trauma resulting in increased activity, decreased activity, or no change in the structures under study. While the authors note and attempt to address this heterogeneity, to some extent this section is problematic because it almost undermines the case they have been making throughout regarding their shared etiology and common pathophysiology. Presumably, underlying these disparate findings, it is reasonable that there must be some coherent mechanism? It might be helpful to at least address this problem.

Response: We have included more commentary and insight to discuss consistent and discrepant findings. While many studies have found different results, commonalities exist that suggest possible common mechanisms. However, we acknowledge this is complicated by several factors, including diagnostic heterogeneity (e.g., trauma exposure vs PTSD, acute substance use vs chronic use or tolerance), use of qualitative phenotypes vs quantitative phenotypes, and small sample size. Ultimately, there is a high degree of phenotypic complexity. Studies also use different imaging modalities or methodology and analysis strategies. We outline ideas to address some of these complexities and lead the field forward.

Overall, I found this review well organized, concise, and well-written. In particular, I appreciated the breadth of material covered, as well as their systematic approach to exploring the clinical presentation, neurobiology, and imaging evidence. I also valued their conclusion, that an approach integrating both disorders is necessary to improve outcomes and find better prevention strategies for this highly vulnerable group.

I believe this review will serve as a useful source for understanding comorbid PTSD and substance use, particularly as it is generalized across substance (most previous reviews have been limited to specific substances).

Thank you for the opportunity to review.

Round 2

Reviewer 1 Report

Dear Authors

Congratulations! You did a very great job!

Good luck!